# Generative Point Cloud Registration

**Haobo Jiang** [1]   **Jin Xie** [2]   **Jian Yang** [3]   **Liang Yu** [4]   **Jianmin Zheng** [1]

## Abstract

In this paper, we propose a novel 3D registration paradigm, *Generative Point Cloud Registration*, which bridges advanced 2D generative models with 3D matching tasks to enhance registration performance. Our key idea is to generate cross-view consistent image pairs that are well-aligned with the source and target point clouds, enabling geometry-color feature fusion to facilitate robust matching. To ensure high-quality matching, the generated image pair should feature both *2D-3D geometric consistency* and *cross-view texture consistency*. To achieve this, we introduce *Match-ControlNet*, a matching-specific, controllable 2D generative model. Specifically, it leverages the depth-conditioned generation capability of ControlNet to produce images that are geometrically aligned with depth maps derived from point clouds, ensuring 2D-3D geometric consistency. Additionally, by incorporating a coupled conditional denoising scheme and coupled prompt guidance, *Match-ControlNet* further promotes cross-view feature interaction, guiding texture consistency generation. Our generative 3D registration paradigm is general and could be seamlessly integrated into various registration methods to enhance their performance. Extensive experiments on 3DMatch and ScanNet datasets verify the effectiveness of our approach. [Code]

## 1. Introduction

Point cloud registration is a problem of finding the optimal rigid transformation, comprising a 3D rotation and a 3D translation, which aligns the source and target point clouds precisely. It plays an important role in various downstream computer vision applications, such as 3D reconstruction, Li-DAR SLAM, and object localization. However, real-world

[1]Nanyang Technological University, Singapore [2]Nanjing University, China [3]Nankai University, China [4]Alibaba Group, China. Correspondence to: Jianmin Zheng <ASJMZheng@ntu.edu.sg>.

*Proceedings of the $42^{nd}$ International Conference on Machine Learning*, Vancouver, Canada. PMLR 267, 2025. Copyright 2025 by the author(s).

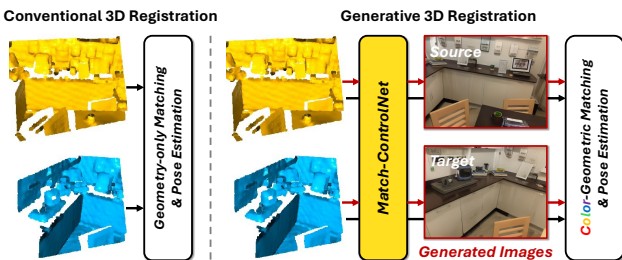

Figure 1. Paradigm comparison of our *generative point cloud registration* with conventional methods. Unlike geometry-only matching in previous methods, our approach introduces *Match-ControlNet*, a matching-specific 2D generative model that generates cross-view images pairs from point cloud data, providing rich color cues for enhanced geometric matching and pose estimation.

challenges like low overlap and noisy points still hinder its adoption in broader real-world scenarios.

Existing 3D registration methods can be roughly categorized into traditional approaches and data-driven deep methods. The traditional approaches include optimization-based fine alignment methods (Besl & McKay, 1992; Yang et al., 2013), which iteratively perform least-squares pose optimization for precise alignment, and handcrafted descriptor-based coarse alignment methods (Rusu et al., 2008; 2009), which capture local geometry to establish correspondences for hypothesize-and-verify registration. Deep registration methods, whether end-to-end (Yew & Lee, 2020; 2022) or descriptor-based (Huang et al., 2021; Qin et al., 2022; Jiang et al., 2023a), exploit the power of deep neural networks to learn discriminative deep 3D features for robust matching. These deep methods significantly enhance the quality of estimated correspondences and improve registration accuracy.

Despite the impressive performance achieved by current point cloud registration methods, their robustness remains limited in challenging scenarios that contain low overlap, repetitive patterns, or noisy points. Recent RGB-D registration studies (Yuan et al., 2023; Mu et al., 2024) have shown that incorporating rich texture and semantic cues from RGB images would significantly enhance the distinctiveness of point cloud descriptors, leading to improved matching accuracy. However, in geometry-only point cloud registration, the RGB images corresponding to the point clouds are unavailable, and existing methods rely solely on 3D geometric information for correspondence estimation and pose calcu-

lation. This raises an interesting question: "*Can we still leverage color information to enhance geometry-only point descriptors for enhanced 3D registration?*"

Motivated by this question and inspired by the recent successes of generative AI models (Ho et al., 2020; Song et al., 2020; Yang et al., 2023; Li et al., 2025; Rombach et al., 2022; Zhang et al., 2023; Jiang et al., 2023b; Wang et al., 2024), we introduce *Generative Point Cloud Registration*, a new 3D matching paradigm that bridges the task gap between the 2D generative models and 3D matching tasks to enhance registration performance. Our key idea is to generate cross-view image pairs that are well-aligned with the corresponding source and target point clouds. These images provide rich color information to complement geometric features, enabling more robust matching (see Fig. 1). Unlike prevalent 2D generative models that focus on single image generation, our matching-specific image generation is pairwise. Importantly, to ensure high matching quality, the generated cross-view image pair should feature two key properties: *2D-3D geometric consistency* and *cross-view texture consistency*. To achieve this, we introduce *Match-ControlNet*, a matching-specific, controllable 2D generative model. *Match-ControlNet* leverages ControlNet's depth-conditioned generation capabilities to produce images geometrically aligned with depth maps (derived from the point cloud pairs), ensuring 2D-3D geometric consistency. Additionally, by incorporating coupled conditional denoising and coupled prompt guidance, *Match-ControlNet* enables effective cross-view image feature interaction, achieving mutual texture message passing and thereby enhancing cross-view texture consistency. Finally, we propose a zero-shot geometric-color fusion mechanism that leverages pretrained large vision models (e.g., DINOv2 (Oquab et al., 2023) and Stable Diffusion (Rombach et al., 2022)) to extract discriminative zero-shot features of generated images for enhancing geometric descriptors via weighted concatenation.

It should be pointed out that our *Generative Point Cloud Registration* framework can operate in both zero-shot and few-shot settings (with minimal fine-tuning samples), each providing valuable color information to enhance precision. Moreover, our framework is general and can be integrated with various 3D registration methods to enhance their matching accuracy. Experiments on 3DMatch and ScanNet datasets validate the effectiveness of our proposed method. To summarize, our contributions are as follows:

- We propose a new *Generative Point Cloud Registration* paradigm, aimed at generating cross-view image pairs for both source and target point clouds, thereby providing rich color information for effective geometric-color feature fusion and improved matching quality.

- Unlike conventional single-image generation, we develop an effective *Match-ControlNet* for matching-specific, pairwise image generation. It incorporates depth-conditioned generation, coupled conditional denoising, and coupled prompt guidance to ensure that the generated image pairs maintain 2D-3D geometric consistency and cross-view texture consistency.

- Our *Generative Point Cloud Registration* framework is general and plug-and-play. Benefiting from our effective *zero-shot geometric-color feature fusion* and *XYZ-RGB fusion* schemes, it can be integrated with various 3D registration approaches to provide free-lunch color information, enhancing their performance.

## 2. Related Work

**Traditional 3D Registration Methods.** Traditional point cloud registration methods are typically categorized into coarse and fine registration approaches. Iterative Closest Point (ICP) (Besl & McKay, 1992), a prominent fine registration method, iteratively computes nearest-neighbor correspondences and performs least-squares optimization for pose estimation. Go-ICP (Yang et al., 2013) enhances ICP's robustness to initialization errors through a branch-and-bound (BnB) global search. Trimmed ICP (Chetverikov et al., 2002) further improves robustness by optimizing over minimal subsets to handle outliers. Additional variants like (Sharp et al., 2002; Fitzgibbon, 2003; Bae & Lichti, 2008; Gressin et al., 2013; Deng et al., 2018) also demonstrate promising precision in fine alignment. Coarse registration methods generally combine handcrafted geometric descriptors with robust pose estimators, such as RANSAC. (Johnson & Hebert, 1999) develops the spin image-based shape descriptors for surface matching and object recognition. USC (Tombari et al., 2010) improves the feature descriptors using an shape context-aware unique local reference frame to improve matching accuracy. SHOT (Salti et al., 2014) introduces a 3D histogram-based feature using normal vectors to describe surface. PFH (Rusu et al., 2008) and FPFH (Rusu et al., 2009) constructs a discriminative and efficient local descriptor based on the oriented histogram with pairwise 3D representations. Other notable coarse methods, including (Mohamad et al., 2014; 2015; Xu et al., 2019; Huang et al., 2017; Ge, 2017), have also achieved impressive registration precision.

**Learning-based Deep Registration Methods.** Deep registration methods primarily consist of end-to-end approaches and deep descriptor-based methods. For end-to-end approaches, DCP (Wang & Solomon, 2019) introduces differentiable soft correspondences for SVD-based pose estimation. RPM-Net (Yew & Lee, 2020) incorporates the Sinkhorn layer and an annealing strategy to mitigate outlier inference. CEMNet and LatentCEM (Jiang et al., 2021a;b) formulate the 3D registration task as a Markov decision process, and introduce a reinforcement learning-driven learning

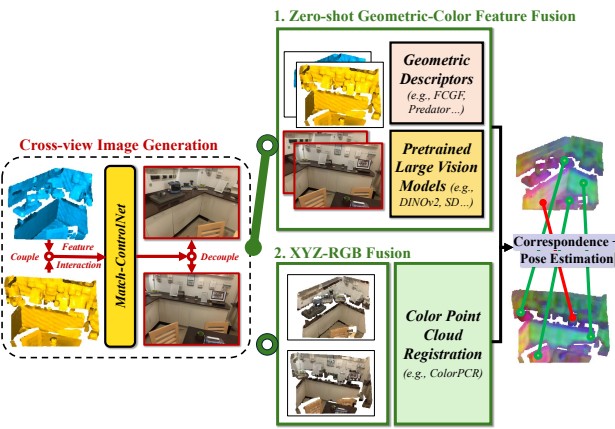

Figure 2. Pipeline of *Generative Point Cloud Registration*. Given a source and a target point cloud, we first apply Match-ControlNet to generate their corresponding images. Next, we employ either zero-shot geometric-color feature fusion or XYZ-RGB fusion to create color-enhanced geometric descriptors, enabling high-quality correspondence estimation and robust pose estimation.

framework for planning-based trial-and-error pose searching (Liu et al., 2024; 2023; Jiang et al., 2021c; 2022). RegTR (Yew & Lee, 2022) designs an effective transformer-based correspondence regression module, addressing large-scale indoor scene registration in an end-to-end manner. For deep descriptor-based methods, 3DMatch (Zeng et al., 2017) employs a Siamese 3D CNN to extract local geometric features for patchwise matching. FCGF (Choy et al., 2019) develops a fully convolutional network to learn dense 3D features for pointwise matching. Predator (Huang et al., 2021) introduces a cross-attention transformer between point cloud pairs for overlap perception and robust registration. Geo-Transformer (Qin et al., 2022) integrates geometric embeddings into the transformer, enhancing feature discrimination. RoITr (Yu et al., 2023) designs a rotation-invariant transformation to further improve the rotational robustness of geometric descriptors. Other methods (Wang et al., 2023; Bai et al., 2020; Li & Harada, 2022; Li et al., 2020; Choy et al., 2020; Chen et al., 2023; Fu et al., 2021; Ao et al., 2023) also demonstrate impressive performance in 3D registration. Beyond traditional and learning-based frameworks, this work introduces a new paradigm: *Generative Point Cloud Registration*. By integrating advanced 2D generative models with the 3D registration domain, our approach generates complementary color information for input point cloud pairs, producing color-enhanced geometric descriptors to improve precision.

## 3. Approach

**Problem Setting.** Given a pair of source and target point clouds $\mathcal{P} = \{\mathbf{p}_i \in \mathbb{R}^3 \mid i = 1, \dots, N\}$ and $\mathcal{Q} = \{\mathbf{q}_i \in \mathbb{R}^3 \mid i = 1, \dots, M\}$, point cloud registration seeks to recover their rigid transformation $\mathbf{T} = \{\mathbf{R}, \mathbf{t}\} \in SE(3)$,

comprising a rotation $\mathbf{R} \in SO(3)$ and a translation $\mathbf{t} \in \mathbb{R}^3$, to align them precisely. The optimal rigid transformation is typically computed by solving:

$$\min_{\mathbf{R},\mathbf{t}} \sum_{(\mathbf{p}^*, \mathbf{q}^*) \in \mathcal{C}^*} \|\mathbf{R} \cdot \mathbf{p}^* + \mathbf{t} - \mathbf{q}^*\|_2^2, \qquad (1)$$

where $\mathcal{C}^*$ denotes the ground-truth correspondences between source and target point clouds. However, $\mathcal{C}^*$ is generally unknown in practical usage, and we need estimate a set of putative correspondences through finding feature nearest neighbor among the pointwise geometric descriptors.

### 3.1. Motivation

Recent RGB-D point cloud registration methods (Yuan et al., 2023; Mu et al., 2024) have demonstrated that RGB images can significantly enhance geometry-only descriptors by providing rich color and semantic information. This enhancement facilitates the construction of higher-quality correspondences, leading to more robust registration.

However, in the context of 3D matching tasks, we focus on geometry-only registration using pure point clouds, as the relevant RGB data is unavailable. To overcome this limitation, we introduce *Generative Point Cloud Registration*, a general framework designed to generate high-quality RGB data for both point clouds, enabling geometric-color feature fusion for enhanced matching. Notably, unlike the conventional single-image generation focused by prevalent 2D generative models (Rombach et al., 2022; Zhang et al., 2023), our matching-specific image generation is pairwise, which should satisfy two key criteria:

**(i) 2D-3D Geometric Consistency:** The generated images should preserve the geometric structure and spatial layout of their respective point clouds to ensure accurate pixel-to-point correspondences and avoid introducing noise;

**(ii) Cross-view Texture Consistency:** The generated image pair should maintain consistent textures for correspondences. Otherwise, inconsistent textures would reduce feature similarity of correspondences, leading to mismatches.

### 3.2. Zero-Shot Geometric Consistency Generation

In this section, we first address the 2D-3D geometric consistency generation through ControlNet (Zhang et al., 2023), a variant of Stable Diffusion (Rombach et al., 2022) with spatially localized image conditions (e.g. Canny edge maps and depth maps).

**Stable Diffusion.** Stable Diffusion is a widely used latent diffusion model for text-to-image generation. It operates within the latent space of a pretrained autoencoder, where a denoiser $\epsilon_\theta(\mathbf{x}_t; t, \mathbf{c})$ (conditioned on the timstamp $t$ and tokenized text prompt $\mathbf{c}$) gradually refines the noisy latent

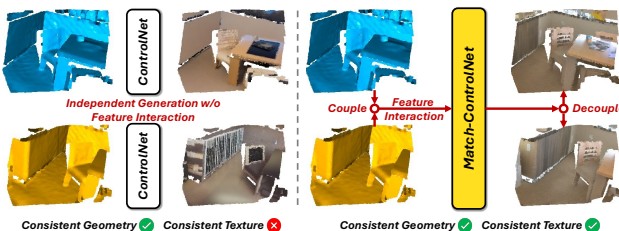

*Figure 3.* Instead of independently performing ControlNet to generate source and target images, our *Match-ControlNet* integrates their denoising generation processes into a unified framework, facilitating feature interaction (i.e., mutual texture message passing) and enhancing their cross-view texture consistency.

feature $\mathbf{x}_t$ to clean one for image decoding. The denoiser follows a UNet architecture with an encoder, middle block, and skip-connected decoder, incorporating stacked transformer and residual modules. Each transformer module utilizes intra-image self-attention for contextual understanding and prompt-to-image cross-attention to guide generation.

**ControlNet-driven 2D-3D Geometric Consistency.** ControlNet further equips the denoiser of Stable Diffusion with a learnable encoder copy for encoding the conditional image $\mathbf{c}_I$, forming a conditional denoiser: $\tilde{\epsilon}_\theta(\mathbf{x}_t; t, \mathbf{c}, \mathbf{c}_I)$. Consequently, the encoded condition features is concatenated with the original encodings of noisy latent representations $\mathbf{x}_t$ for conditional feature decoding via skip connections. Notably, ControlNet allows the use of depth maps as conditional inputs to generate RGB images that preserve geometric structures well-aligned with the provided depth prior. This capability perfectly aligns with our objective and motivates us to convert the source and target point clouds into their corresponding depth maps, $\mathbf{D}_\mathcal{P}$ and $\mathbf{D}_\mathcal{Q} \in \mathbb{R}^{H \times W \times 1}$, via the intrinsic matrix. Then, each produced depth map can be independently used to condition ControlNet to produce the geometrically consistent image.

### 3.3. Zero-Shot Texture Consistency Generation

Although the original ControlNet can generate source and target image pairs that are geometrically well-aligned with the given point cloud pair, the texture details of the corresponding regions between the generated image pair often differ as shown in Fig. 3 (*left*). This texture inconsistency primarily arises because the denoising processes for the source and target images operate independently as follows:

$$\begin{aligned} \tilde{\epsilon}_\theta(\mathbf{x}_t^\mathcal{P}; t, \mathbf{c}, \mathbf{d}_\mathcal{P}) \to \mathbf{x}_{t-1}^\mathcal{P}, \\ \tilde{\epsilon}_\theta(\mathbf{x}_t^\mathcal{Q}; t, \mathbf{c}, \mathbf{d}_\mathcal{Q}) \to \mathbf{x}_{t-1}^\mathcal{Q}, \end{aligned} \quad (2)$$

with each unaware of the colors produced by the other. Here, $\mathbf{x}_t^\mathcal{P}, \mathbf{x}_t^\mathcal{Q} \in \mathbb{R}^{H' \times W' \times d}$ denote the noisy latent representations corresponding to source and target images; $\mathbf{d}_\mathcal{P}, \mathbf{d}_\mathcal{Q} \in \mathbb{R}^{H' \times W' \times d}$ represent the encoded features of depth maps $\mathbf{D}_\mathcal{P}$ and $\mathbf{D}_\mathcal{Q}$ via optimized zero convolutions of ControlNet. This insight motivates us to combine source

and target image denoising generation processes into a joint denoising pass, thereby enabling mutual texture message passing and promoting texture consistency (see Fig. 3).

Based on this motivation, we establish *Match-ControlNet*, an improved ControlNet variant for matching-specific, conditional image generation. Following Sec. 3.2, we still take depth maps derived by point clouds as conditional images so as to inherit ControlNet's 2D-3D geometric consistency generation capability. Additionally, we introduce two key designs: coupled conditional denoising and coupled prompt guidance to achieve the cross-view texture consistency generation. The details of these two designs are as follows:

**Coupled Conditional Denoising.** To achieve mutual texture message passing, a straightforward approach is to build two denoisers and incorporate an additional cross-denoiser attention module to facilitate their message passing. However, running two denoisers simultaneously is inefficient, and such significant architectural changes would require extensive model fine-tuning.

To enable effective cross-view message passing without any finetuning (i.e., zero-shot), we propose an efficient coupled conditional denoising scheme for joint, interactive source and target image generations. Specifically, we expand the noisy latent representation $\mathbf{x}_t^{\mathcal{P}(\mathcal{Q})}$ with shape $[H', W', d]$ to a coupled one $\mathbf{x}_t^{\mathcal{P}\mathcal{Q}}$ with the extended shape $[2H', W', d]$. Also, we vertically concatenate the source and target depth maps into a coupled one $\mathbf{D}_{\mathcal{P}\mathcal{Q}} \in \mathbb{R}^{2H \times W \times 1}$, and further emply the ControlNet's zero convolutions to encode it into the condition features $\mathbf{d}_{\mathcal{P}\mathcal{Q}} \in \mathbb{R}^{2H' \times W' \times d}$. Consequently, without any architectural modifications or parameter fine-tuning, the original conditional denoiser can be directly employed for our coupled conditional denoising:

$$\tilde{\epsilon}_\theta\left(\mathbf{x}_t^{\mathcal{P}\mathcal{Q}}; t, \mathbf{c}, \mathbf{d}_{\mathcal{P}\mathcal{Q}}\right) \to \mathbf{x}_{t-1}^{\mathcal{P}\mathcal{Q}}, \quad (3)$$

forming the denoising Markov chain: $\mathbf{x}_T^{\mathcal{P}\mathcal{Q}} \to \cdots \mathbf{x}_1^{\mathcal{P}\mathcal{Q}} \to \mathbf{x}_0^{\mathcal{P}\mathcal{Q}}$. Here, the initial coupled latent representation $\mathbf{x}_T^{\mathcal{P}\mathcal{Q}}$ is sampled from a standard Gaussian distribution $\mathcal{N}(\mathbf{0}, \mathbf{I})$. To further clarify the cross-view texture message passing during our coupled denoising process, we formulate the self-attention mechanism within the denoiser as $SA(\mathbf{x}_t^{\mathcal{P}\mathcal{Q}}) =$

$$\text{softmax}\left(\frac{(\mathbf{x}_t^{\mathcal{P}\mathcal{Q}}\mathbf{W}^q)(\mathbf{x}_t^{\mathcal{P}\mathcal{Q}}\mathbf{W}^k)^\top}{\sqrt{d}}\right)(\mathbf{x}_t^{\mathcal{P}\mathcal{Q}}\mathbf{W}^v), \quad (4)$$

where $\mathbf{W}^q$, $\mathbf{W}^k$ and $\mathbf{W}^v \in \mathbb{R}^{d \times d}$ denote the projection matrices for queries, keys and values, respectively. Eq. 4 illustrates that by coupling the source and target noisy latent representations, each feature element can establish long-range dependencies with all feature elements from both the source and target feature maps, allowing effective cross-view feature interaction and texture-aware message passing for promoting texture consistency generation.

**Coupled Prompt Guidance.** Although the aforementioned coupled denoising generative mechanism has provided the essential components for texture consistency generation, the denoiser still fails to produce the image pair with expected cross-view texture consistency. The core reason is that the denoiser does not know what kind of image the user expects to generate, and we need to tell it what to do. Our important finding is that when we use a specific prompt (named *coupled prompt*) as below to tell our *Match-ControlNet* to produce the vertically stacked images with consistent layout and elements:

"*Generate two vertically stacked images that are captured from the different viewpoints in a same scene. The images should feature the same environment—whether indoor or outdoor, like a living room, office, street, or natural landscape—with very subtle differences between them. Overall, the layout and key elements remain the same.*",

the denoiser can naturally be guided to recover consistent textures without any model fine-tuning. To the best of our knowledge, we are the first to uncover and utilize this inherent capability of the pre-trained ControlNet for zero-shot pairwise image generation enjoying both 2D-3D geometric consistency and cross-view texture consistency.

### 3.4. Few-Shot Consistency Fine-tuning

Although our zero-shot *Match-ControlNet* above has demonstrated promising pairwise consistency generation capabilities, Fig. 4 shows that some corresponding regions may still exhibit geometric or texture inconsistency issues. To mitigate it, we further propose a few-shot finetuning mechanism to improve the consistency generation quality of our *Match-ControlNet*. It's noted that we only finetune the learnable encoder copy of *Match-ControlNet* rather than the all parameters to preserve the powerful generation ability of Stable Diffusion. Specifically, we first collect a set of RGB-depth pairs, $\{((\mathbf{I}_{\mathcal{P}}, \mathbf{D}_{\mathcal{P}}), (\mathbf{I}_{\mathcal{Q}}, \mathbf{D}_{\mathcal{Q}}))_j\}$ ($j$ denotes the sample index), as the training data. For each sample pair, we then concatenate its depth maps and RGB images into coupled ones $\{(\mathbf{I}_{\mathcal{PQ}}, \mathbf{D}_{\mathcal{PQ}})_j\}$. Finally, we use the loss function below to finetune the denoiser:

$$\mathcal{L} = \mathbb{E}_{\mathbf{x}_t^{\mathcal{PQ}}, t, \tilde{\mathbf{c}}, \mathbf{d}_{\mathcal{PQ}}, \epsilon \sim \mathcal{N}(0,1)} \left[ \left\| \epsilon - \tilde{\epsilon}_\theta(\mathbf{x}_t^{\mathcal{PQ}}, t, \tilde{\mathbf{c}}, \mathbf{d}_{\mathcal{PQ}}) \right\|_2^2 \right],$$
$$(5)$$

where $\mathbf{x}_t^{\mathcal{PQ}}$ represents the diffused latent representation of the coupled image $\mathbf{I}_{\mathcal{PQ}}$; $\tilde{\mathbf{c}}$ denotes the token sequence of our *coupled prompt*. Our experiments show that even with a limited number of samples ($\sim$3K), few-shot finetuning can effectively improves the quality of consistency generation.

### 3.5. Geometric-Color Fused Point Descriptor

In this section, we focus on how to enhance the geometric representations of point clouds with the free-lunch color

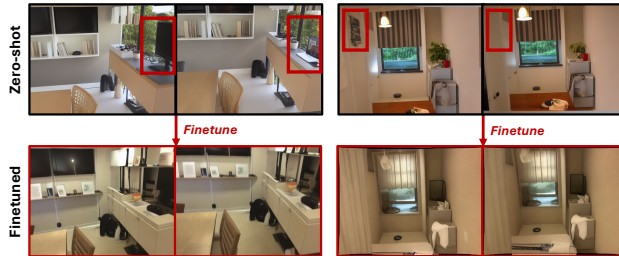

*Figure 4.* Compared to the *zero-shot Match-ControlNet* (top), the *finetuned Match-ControlNet* can tend to achieve higher 2D-3D geometric consistency and the cross-view texture consistency.

information from generated source and target images, $\tilde{\mathbf{I}}_{\mathcal{P}}$ and $\tilde{\mathbf{I}}_{\mathcal{Q}}$. We provide two geometric-color fusion schemes:

**Zero-Shot Geometric-Color Feature Fusion.** Inspired by the powerful RGB representations of large vision models, we utilize them to directly extract zero-shot semantic features from the generated images. Specifically, we employ two widely-used vision foundation models: DINOv2 (Oquab et al., 2023) and Stable Diffusion (Rombach et al., 2022) for image encoding, achieving corresponding feature maps. These feature maps are then projected into the point cloud space using the camera intrinsic matrix, yielding pointwise color descriptors: $\{\mathbf{f}_{\mathbf{p}_i}^{rgb}\}$ and $\{\mathbf{f}_{\mathbf{q}_i}^{rgb}\}$ for both source and target point clouds. Finally, we apply a simple weighted concatenation to combine the RGB descriptors with the geometric descriptors, resulting in fused descriptors as follows:

$$\tilde{\mathbf{f}}_{\mathbf{P}_i} = [\omega \cdot \mathbf{f}_{\mathbf{P}_i}^{geo}; (1-\omega) \cdot \mathrm{PCA}(\mathbf{f}_{\mathbf{P}_i}^{rgb})] \in \mathbb{R}^{d_{geo}+d_{rgb}}, \quad (6)$$

where $[\cdot; \cdot]$ denotes the feature concatenation operator; $\omega$ is the fusion weight and $\mathbf{f}_{\mathbf{P}_i}^{geo}$ represents the geometric descriptors; $\mathrm{PCA}(\cdot)$ denotes the principal component analysis function to compress the feature dimension of the color descriptor to fit that of the geometric descriptor. The same fusion scheme is also applied to the target point clouds. Notably, this zero-shot geometric-color fusion approach is general and can be applied to a variety of geometric descriptors, whether traditional or deep descriptors.

**XYZ-RGB Fusion.** This fusion scheme directly projects the generated source and target RGB images into the point cloud space. The resulting RGB values are then concatenated with the point coordinates of the point clouds, forming 6D color source and target point clouds, as shown in Fig. 5 (*left*). These color point clouds are subsequently used as inputs to the color point cloud registration method, like ColorPCR (Mu et al., 2024), for 3D registration.

## 4. Experiments

### 4.1. Experimental Setting

**Implementation Details.** During the few-shot fine-tuning stage, we randomly select 3,000 sample pairs from the Scan-

*Table 1.* Comparison of the methods on rotation, translation, and Chamfer distance on **ScanNet** (Dai et al., 2017) benchmark dataset.

| | Rotation (deg) | | | | | Translation (cm) | | | | | Chamfer (mm) | | | | |
|---|---|---|---|---|---|---|---|---|---|---|---|---|---|---|---|
| | Accuracy ↑ | | | Error↓ | | Accuracy ↑ | | | Error↓ | | Accuracy ↑ | | | Error↓ | |
| **Methods** | 5 | 10 | 45 | Mean | Med. | 5 | 10 | 25 | Mean | Med. | 1 | 5 | 10 | Mean | Med. |
| FPFH (Rusu et al., 2009) | 41.4 | 56.7 | 73.3 | 39.2 | 7.1 | 17.5 | 35.1 | 50.9 | 79.5 | 23.5 | 32.3 | 48.0 | 53.0 | 159.6 | 6.5 |
| Lepard (Li & Harada, 2022) | 63.3 | 75.5 | 84.1 | 24.9 | 3.3 | 31.3 | 56.4 | 72.3 | 48.4 | 8.1 | 51.6 | 69.1 | 73.0 | 89.2 | 0.9 |
| RegTR (Yew & Lee, 2022) | 72.5 | 83.8 | 94.1 | 10.2 | 2.3 | 44.3 | 65.6 | 80.0 | 27.7 | 5.8 | 61.0 | 76.6 | 80.9 | 54.0 | 0.5 |
| RoITr (Yu et al., 2023) | 70.0 | 77.5 | 83.7 | 24.1 | 2.3 | 40.3 | 62.3 | 75.1 | 45.6 | 6.5 | 58.8 | 72.4 | 75.4 | 94.1 | 0.6 |
| FCGF (Choy et al., 2019) | 78.9 | 84.2 | 87.5 | 19.4 | 1.5 | 55.3 | 70.7 | 79.7 | 37.8 | 4.3 | 67.3 | 78.2 | 80.3 | 100.7 | 0.4 |
| Generative FCGF$^{\text{DINOv2}}$ | 81.0 | 86.2 | 89.4 | 16.5 | 1.4 | **57.3** | 72.6 | 80.9 | 33.9 | **4.0** | **68.9** | 79.5 | 81.5 | 97.4 | **0.3** |
| Generative FCGF$^{\text{SD}}$ | **82.9** | **90.0** | **94.4** | **8.4** | 1.6 | 56.4 | **73.0** | **82.7** | **21.7** | 4.1 | 67.7 | **80.9** | **83.7** | **66.0** | 0.4 |
| *Improvement* ↑ | *4.0* | *5.8* | *6.9* | *11.0* | *0.1* | *2.0* | *2.3* | *3.0* | *16.1* | *0.3* | *1.6* | *2.7* | *3.4* | *34.7* | *0.1* |
| Predator (Huang et al., 2021) | 64.3 | 75.2 | 82.6 | 26.3 | 3.2 | 30.1 | 54.8 | 69.2 | 48.7 | 8.4 | 50.8 | 66.9 | 70.6 | 93.2 | 1.0 |
| Generative Predator$^{\text{DINOv2}}$ | 67.0 | 78.0 | 87.2 | 19.0 | 3.0 | 30.7 | 56.0 | 70.3 | 41.4 | 8.1 | 52.0 | 67.8 | 71.3 | 79.2 | 0.9 |
| Generative Predator$^{\text{SD}}$ | **70.7** | **81.3** | **88.7** | **17.0** | **2.8** | **33.0** | **59.4** | **73.3** | **36.6** | **7.5** | **54.7** | **70.8** | **74.2** | **72.5** | **0.8** |
| *Improvement* ↑ | *6.4* | *6.1* | *6.1* | *9.3* | *0.4* | *2.9* | *4.6* | *4.1* | *12.1* | *0.9* | *3.9* | *3.9* | *3.6* | *20.7* | *0.2* |
| GeoTrans (Qin et al., 2022) | 71.5 | 78.0 | 83.4 | 26.2 | 2.0 | 48.4 | 65.2 | 74.6 | 51.9 | 5.2 | 62.0 | 72.5 | 75.0 | 97.3 | 0.5 |
| Generative GeoTrans$^{\text{DINOv2}}$ | 74.3 | 81.0 | 87.6 | 19.7 | 1.9 | 50.8 | 67.4 | 76.0 | 41.8 | 4.9 | 63.7 | 73.9 | 76.2 | 86.2 | **0.4** |
| Generative GeoTrans$^{\text{SD}}$ | **77.2** | **84.0** | **89.9** | **16.5** | **1.8** | **51.3** | **68.7** | **78.4** | **35.6** | **4.8** | **65.2** | **76.1** | **78.7** | **71.0** | **0.4** |
| *Improvement* ↑ | *5.7* | *6.0* | *6.5* | *9.7* | *0.2* | *2.9* | *3.5* | *3.8* | *16.3* | *0.4* | *3.2* | *3.6* | *3.7* | *26.3* | *0.1* |

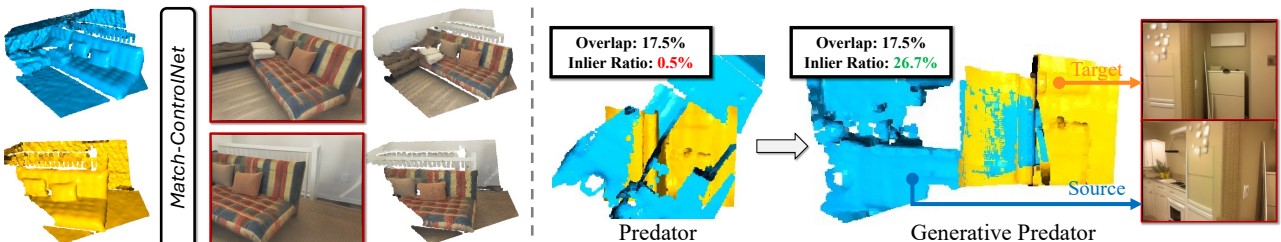

*Figure 5. **Left:*** The visualization of the generated RGB image pairs and the formed color source and target point clouds; ***Right:*** In low-overlap cases, the original Predator struggles with registration. By contrast, the *Generative Predator*, enhanced with generated color information, successfully align them well.

Net training set (Dai et al., 2017) for model fine-tuning. Following the default fine-tuning configuration of ControlNet (Zhang et al., 2023), we adopt the AdamW optimizer (Loshchilov, 2017) with a learning rate of 1e-5 and set the training epoch to 10. The code for this project is implemented in PyTorch, and all experiments are conducted on a server equipped with an Intel i5 2.2 GHz CPU and a TITAN RTX GPU. In our experiments, we integrate our zero-shot geometric-color fusion (Sec. 3.5) with three prevalent deep geometric descriptors: FCGF (Choy et al., 2019), Predator (Huang et al., 2021), and GeoTransformer (Qin et al., 2022), resulting in corresponding color-enhanced variants: **Generative FCGF**, **Generative Predator**, and **Generative GeoTrans** for method evaluation. Additionally, to validate our XYZ-RGB fusion scheme, we replace the real color point clouds (with actual RGB values) used by ColorPCR (Mu et al., 2024) with our generated color point clouds (with synthesized RGB values), forming **Generative ColorPCR** for 3D matching.

**Evaluation Metric.** Following (El Banani et al., 2021; Yuan et al., 2023), we use rotation error, translation error, and Chamfer error, including the accuracy across varying thresholds and mean/median errors, for performance evaluation.

## 4.2. Comparison with Existing Methods

**Evaluation on ScanNet.** We first perform model evaluation on a widely-used, large-scale indoor benchmark dataset, ScanNet (Dai et al., 2017). We follow the official data split to divide this dataset into the training, validation, and testing subsets, and construct view pairs by sampling image pairs that are 50 frames apart. Compared to the 20-frame separation used in (El Banani et al., 2021; Yuan et al., 2023), our approach with a 50-frame separation further reduces the overlap ratio (i.e., lower overlap), thereby increasing the registration difficulty. We compare our method against with one representative traditional descriptor: FPFH (Rusu et al., 2009), one scene-level end-to-end registration network: RegTR (Yew & Lee, 2022), and five deep descriptors: FCGF (Choy et al., 2019), Predator (Huang et al., 2021), GeoTrans (Qin et al., 2022), Lepard (Li & Harada, 2022), and RoITr (Yu et al., 2023). We adopt RANSAC-50k as the pose estimator for FPFH, Lepard and RoITr, and select

*Table 2.* Comparison of the methods on rotation, translation, and Chamfer distance on **3DMatch** (Zeng et al., 2017) benchmark dataset.

| Methods | Rotation (deg) | | | | | Translation (cm) | | | | | Chamfer (mm) | | | | |
| | Accuracy ↑ | | | Error↓ | | Accuracy ↑ | | | Error↓ | | Accuracy ↑ | | | Error↓ | |
| | 5 | 10 | 45 | Mean | Med. | 5 | 10 | 25 | Mean | Med. | 1 | 5 | 10 | Mean | Med. |
|---|---|---|---|---|---|---|---|---|---|---|---|---|---|---|---|
| FPFH (Rusu et al., 2009) | 69.1 | 82.9 | 91.2 | 15.0 | 3.1 | 25.8 | 53.9 | 75.1 | 37.4 | 9.1 | 52.5 | 74.2 | 79.2 | 57.6 | 0.9 |
| Lepard (Li & Harada, 2022) | 84.3 | 91.0 | 94.1 | 11.1 | 2.1 | 43.1 | 75.2 | 88.9 | 21.8 | 5.8 | 72.1 | 88.3 | 90.5 | 45.3 | 0.4 |
| RegTR (Yew & Lee, 2022) | 86.2 | 92.1 | 97.2 | 5.7 | 1.6 | 55.0 | 77.6 | 88.9 | 18.8 | 4.6 | 75.4 | 88.2 | 91.3 | 40.0 | 0.3 |
| RoITr (Yu et al., 2023) | 86.3 | 91.1 | 93.8 | 11.1 | 1.6 | 51.2 | 77.4 | 89.1 | 20.5 | 4.9 | 75.2 | 88.5 | 90.6 | 50.1 | 0.4 |
| FCGF (Choy et al., 2019) | 90.4 | 93.7 | 94.8 | 9.4 | **1.4** | 53.4 | 79.3 | 91.0 | 19.2 | 4.7 | 76.7 | 90.8 | 92.4 | 40.3 | **0.4** |
| Generative FCGF$^{DINOv2}$ | 91.5 | 94.3 | 95.3 | 8.5 | **1.4** | 53.6 | 79.3 | 91.5 | 18.1 | **4.6** | 77.5 | 91.1 | 92.7 | 41.1 | **0.4** |
| Generative FCGF$^{SD}$ | **94.3** | **96.7** | **98.1** | **4.5** | **1.4** | **54.3** | **81.5** | **93.1** | **12.5** | 4.7 | **78.2** | **92.9** | **94.6** | **37.7** | **0.4** |
| *Improvement* ↑ | *3.9* | *3.0* | *3.3* | *4.9* | *0.0* | *0.9* | *2.2* | *2.1* | *6.7* | *0.0* | *1.5* | *2.1* | *2.2* | *2.6* | *0.0* |
| Predator (Huang et al., 2021) | 85.0 | 91.5 | 94.2 | 10.5 | 2.0 | 42.1 | 72.5 | 87.1 | 22.6 | 5.8 | 71.2 | 85.8 | 88.6 | 45.0 | 0.5 |
| Generative Predator$^{DINOv2}$ | 88.1 | **94.8** | 96.9 | 6.2 | **1.8** | 44.7 | 73.9 | 88.4 | **15.5** | 5.6 | 72.4 | 87.7 | 90.8 | **33.1** | 0.4 |
| Generative Predator$^{SD}$ | **88.6** | 94.6 | **97.0** | **5.9** | 1.9 | **45.7** | **74.5** | **89.1** | 15.7 | **5.5** | **73.3** | **88.3** | **90.9** | 40.4 | **0.4** |
| *Improvement* ↑ | *3.6* | *3.1* | *2.8* | *4.6* | *0.2* | *3.6* | *2.0* | *2.0* | *7.1* | *0.3* | *2.1* | *2.5* | *2.3* | *11.9* | *0.1* |
| GeoTrans (Qin et al., 2022) | 88.9 | 91.8 | 93.3 | 12.0 | 1.4 | 59.8 | 81.0 | 90.1 | 24.6 | 4.0 | 79.2 | 89.0 | 90.6 | 53.3 | **0.3** |
| Generative GeoTrans$^{DINOv2}$ | 90.2 | 93.2 | 95.2 | 8.9 | 1.4 | 61.0 | **83.1** | 90.4 | **16.9** | 3.9 | 80.4 | 89.7 | 91.7 | **36.9** | **0.3** |
| Generative GeoTrans$^{SD}$ | **91.5** | **94.3** | **96.2** | **7.6** | 1.4 | **61.3** | 82.9 | **90.9** | 17.2 | 3.9 | **81.5** | **90.1** | **92.3** | 37.3 | **0.3** |
| *Improvement* ↑ | *2.6* | *2.5* | *2.9* | *4.4* | *0.0* | *1.5* | *2.1* | *0.8* | *7.7* | *0.1* | *2.3* | *1.1* | *1.7* | *16.4* | *0.0* |

SC2PCR (Chen et al., 2022), RANSAC-50k and LGR for (Generative) FCGF, (Generative) Predator, and (Generative) GeoTrans (Qin et al., 2022), to validate the robustness of our generative 3D registration paradigm across different pose estimators. Table 1 demonstrates that enhanced by the free-lunch color information generated by our *Match-ControlNet*, all generative versions of FCGF, Predator, and GeoTrans achieve significant performance improvements, such as 6.9% ↑ of Generative FCGF on 45°@Rotation metric. These confirm the generality and effectiveness of our proposed *generative point cloud registration* paradigm. Additionally, we find that compared to the DINOv2 image encoding, Stable Diffusion can capture more discriminative representations and achieve higher precisions.

**Evaluation on 3DMatch.** We next evaluate our method on 3DMatch (Zeng et al., 2017), another widely-used benchmark dataset for 3D registration. We follow (El Banani et al., 2021; Yuan et al., 2023) as in ScanNet to produce the pairwise samples. Also, we increase the view separation from 20 to 40, resulting in point cloud pairs with lower overlap to increase the registration challenge. Table. 2 demonstrates that by incorporating FCGF, Predator, and GeoTrans into our generative point cloud registration framework, their generative variants also consistently achieve the performance gain, validating the effectiveness of our proposed paradigm.

### 4.3. Ablation Studies and Analysis

**Effectiveness of Match-ControlNet.** We first evaluate the performance contribution of our developed *Match-ControlNet*: **(i)** The top block of Table 3 demonstrates that, compared to using generated image pairs with only 2D-3D geometric consistency (*geo*), incorporating both 2D-3D

geometric consistency and cross-view texture consistency (*geo+tex*) through our *Match-ControlNet* results in higher registration accuracy. This improvement is due to the additional benefit of consistent textures and colors, which further facilitate accurate correspondence identification. Additionally, we observe that the generated images with only 2D-3D geometric consistency can also bring performance gain in some criteria. We attribute it to that DINOv2 and Stable Diffusion can extract powerful semantic representations, mitigating the feature inconsistency of correspondences caused by the texture difference and thereby aiding correspondence identification. Furthermore, we visualize the generated image pair for given source and target point clouds in Fig. 5 (*left*). It shows that our *Match-ControlNet* is capable of producing high-quality image pairs with consistent 2D-3D geometry and cross-view texture.

**Zero-Shot vs Finetuning.** We further investigate the performance of *Match-ControlNet* in both zero-shot and finetuned settings. As shown in the second block of Table 3, both approaches yield substantial improvements over FCGF. Moreover, because the finetuned *Match-ControlNet* benefits from task-specific training, it consistently achieves higher registration accuracy than the zero-shot version. Notably, even few-shot finetuning with as few as 1K samples yields clear performance gains. Increasing the number of finetuning samples (e.g., to 3K or 5K) provides additional improvements; however, models trained on 3K or 5K samples show comparable registration accuracy in practice. Hence, we adopt 3K samples as our default finetuning configuration.

**Zero-Shot Geometric-Color Feature Fusion.** We next conduct ablation studies on the zero-shot geometric-color feature fusion described in Eq. 6. As shown in the fourth block of Table 3, Generative GeoTrans exhibits varying

Table 3. Ablation studies on **3DMatch** (Zeng et al., 2017) dataset. (*) denotes the default configuration.

| Methods | Rotation (deg) Accuracy↑ 5 | 10 | 45 | Error↓ Mean | Med. | Translation (cm) Accuracy↑ 5 | 10 | 25 | Error↓ Mean | Med. | Chamfer (mm) Accuracy↑ 1 | 5 | 10 | Error↓ Mean | Med. |
|---|---|---|---|---|---|---|---|---|---|---|---|---|---|---|---|
| FCGF | 90.4 | 93.7 | 94.8 | 9.4 | **1.4** | 53.4 | 79.3 | 91.0 | 19.2 | **4.7** | 76.7 | 90.8 | 92.4 | 40.3 | **0.4** |
| Generative FCGF$^{SD}$ (geo) | 92.4 | 96.1 | 97.8 | 5.2 | 1.5 | 53.3 | 79.2 | 91.7 | 13.5 | 4.8 | 75.6 | 91.1 | 93.1 | **35.8** | **0.4** |
| Generative FCGF$^{SD}$ (geo + tex) | **94.3** | **96.7** | **98.1** | **4.5** | 1.4 | 54.3 | 81.5 | 93.1 | 12.5 | 4.7 | 78.2 | 92.9 | 94.6 | 37.7 | **0.4** |
| Generative FCGF$^{SD}$ (zero-shot) | 92.4 | 96.1 | 97.3 | 5.4 | 1.5 | **54.3** | 80.3 | 92.4 | 13.0 | **4.6** | 77.3 | 92.1 | 94.0 | **33.7** | **0.4** |
| Generative FCGF$^{SD}$ (finetuning) | 94.3 | 96.7 | 98.1 | 4.5 | 1.4 | 54.3 | 81.5 | 93.1 | 12.5 | 4.7 | 78.2 | 92.9 | 94.6 | 37.7 | **0.4** |
| Finetune (#samples=1000) | 93.5 | 96.8 | 98.1 | 4.6 | **1.4** | 54.2 | 80.5 | 92.5 | 12.4 | 4.7 | **78.2** | 92.5 | 94.7 | 32.8 | **0.4** |
| Finetune (#samples=3000)* | 94.3 | 96.7 | 98.1 | 4.5 | 1.4 | 54.3 | 81.5 | 93.1 | 12.5 | 4.7 | 78.2 | 92.9 | 94.6 | 37.7 | **0.4** |
| Finetune (#samples=5000) | 93.6 | **97.2** | 98.0 | **4.4** | 1.5 | 54.0 | 80.8 | 92.7 | **11.9** | **4.5** | 77.7 | 92.9 | 94.2 | **32.3** | **0.4** |
| ColorPCR (Mu et al., 2024) | 79.9 | 84.6 | 88.9 | 16.5 | **1.8** | **48.3** | 69.6 | 82.2 | 41.8 | **5.2** | 66.6 | 80.6 | 83.3 | 81.1 | 0.5 |
| Generative ColorPCR | **83.6** | **89.8** | **93.2** | **12.0** | 1.9 | 47.3 | **73.3** | **86.7** | **28.3** | 5.3 | **70.3** | **85.7** | **88.2** | **59.1** | **0.4** |
| *Improvement↑* | *3.7* | *5.2* | *4.3* | *4.5* | *0.1* | *1.0* | *3.7* | *4.5* | *13.5* | *0.1* | *3.7* | *5.1* | *4.9* | *22.0* | *0.1* |
| Color feat. dim. $d_{rgb} = 16$ | 90.8 | 93.9 | **96.3** | **7.6** | 1.4 | **62.2** | **83.4** | 90.7 | 18.1 | 3.9 | 81.5 | 89.7 | 91.7 | 41.8 | **0.3** |
| Color feat. dim. $d_{rgb} = 32$ | **91.5** | **94.3** | 96.2 | **7.6** | 1.4 | 61.3 | 82.9 | 90.9 | **17.2** | 3.9 | 81.5 | 90.1 | 92.3 | **37.3** | **0.3** |
| Color feat. dim. $d_{rgb} = 64$* | **91.5** | **94.3** | 96.2 | **7.6** | 1.4 | 61.3 | 82.9 | 90.9 | **17.2** | 3.9 | 81.5 | 90.1 | 92.3 | **37.3** | **0.3** |
| Color feat. dim. $d_{rgb} = 128$ | 91.0 | **94.3** | 96.0 | 8.1 | **1.4** | 62.0 | **83.4** | **91.6** | 18.1 | **3.8** | 81.5 | **90.4** | **92.4** | 41.2 | **0.3** |
| Fusion weight $\omega = 0.0$ | 88.3 | 93.3 | 96.6 | 6.9 | 1.6 | 53.7 | 77.0 | 86.5 | 20.8 | 4.7 | 74.6 | 85.9 | 88.5 | 50.5 | 0.4 |
| Fusion weight $\omega = 0.25$ | 90.7 | **95.5** | **97.2** | **6.1** | 1.5 | 57.9 | 81.5 | 89.8 | **16.5** | 4.2 | 78.6 | 89.2 | 92.0 | 40.4 | **0.3** |
| Fusion weight $\omega = 0.50$* | **91.5** | 94.3 | 96.2 | 7.6 | **1.4** | **61.3** | **82.9** | **90.9** | 17.2 | **3.9** | **81.5** | **90.1** | **92.3** | **37.3** | **0.3** |
| Fusion weight $\omega = 0.75$ | 89.2 | 92.7 | 93.9 | 10.9 | **1.4** | 60.5 | 81.7 | 90.6 | 22.6 | 4.0 | 79.5 | 89.3 | 91.4 | 49.5 | **0.3** |
| Fusion weight $\omega = 1.0$ | 89.0 | 91.8 | 93.3 | 12.0 | **1.4** | 59.9 | 81.1 | 90.2 | 24.6 | 4.0 | 79.3 | 89.1 | 90.8 | 53.3 | **0.3** |

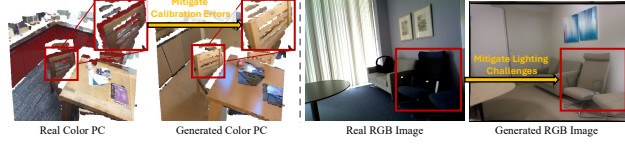

Real Color PC    Generated Color PC    Real RGB Image    Generated RGB Image

*Figure 6.* Our *Match-ControlNet* effectively mitigates calibration errors and lighting challenges commonly encountered in real-world RGB-D data, thereby improving the matching precision of color point cloud registration methods.

registration performance under different color feature dimensions, $d_{rgb} \in \{16, 32, 64, 128\}$. We observe that a very small color feature dimension (e.g., $d_{rgb} = 16$) degrades performance due to limited semantic representational capacity, while excessively large dimensions do not yield significant performance gains. Therefore, to balance inference efficiency with registration precision, we set $d_{rgb} = 64$ as our default setting. Additionally, in the fifth block of Table 3, we investigate performance variations with different fusion weights $\omega \in \{0.0, 0.25, 0.50, 0.75, 1.0\}$, where a larger $\omega$ places more emphasis on the geometric descriptors (see Eq. 6). Our results indicate that both overly high $\omega$ (which overemphasizes geometry) and overly low $\omega$ (which overemphasizes color) lead to degraded registration accuracy. By contrast, a balanced weight (e.g., $\omega = 0.50$) achieves higher performance. As a result, we adopt $\omega = 0.50$ as our default hyperparameter configuration.

**XYZ-RGB Fusion.** We finally evaluate the effectiveness of the XYZ-RGB fusion (see Sec. 3.5). The third block in Table 3 demonstrates that, on the 3DMatch dataset, Genera-

tive ColorPCR with the synthesized color even outperforms the original ColorPCR with the real color. This advantage is attributed not only to the high-quality pairwise image generation provided by our *Match-ControlNet*, but also to several key benefits of our generated XYZ-RGB data over real XYZ-RGB data: **(i)** Mitigating calibration errors: As shown in Fig. 6 (*left*), some real RGB-D data would suffer from calibration errors, which may lead to misalignment in the colored point clouds. By contrast, our framework, benefiting from the powerful 2D-3D consistency generation ability, effectively reduces such calibration errors, producing higher-quality colored point clouds and enabling more accurate matching. **(ii)** Mitigating lighting challenges: Fig. 6 (*right*) shows that RGB data from real-world conditions can degrade under poor lighting, negatively impacting RGB-D matching performance. Our generative point cloud registration framework, however, can generate images with consistent lighting conditions, independent of real-world lighting issues, thus enhancing the overall lighting robustness.

## 5. Conclusion

We have introduced a novel 3D registration paradigm, *generative point cloud registration*, which effectively leverages advanced 2D generative models to augment geometry-only 3D registration. To this end, we developed *Match-ControlNet*, a matching-specific variant of ControlNet designed to synthesize paired RGB images for both source and target point clouds. By integrating depth-conditioned generation from ControlNet, coupled conditional denoising,

and coupled prompt guidance, these generated RGB image pairs preserve both 2D-3D geometric consistency and cross-view texture consistency, thereby facilitating high-quality 3D matching. Notably, our generative framework is general and can be incorporated into a variety of registration methods to improve their performance. Extensive experiments demonstrate the effectiveness of the proposed paradigm.

## Impact Statement

This paper presents work whose goal is to advance research in point cloud registration. While potential societal implications may exist, we do not find any that necessitate explicit discussion here.

## Acknowledgments

This research is supported by the RIE2025 Industry Alignment Fund – Industry Collaboration Projects (IAF-ICP) (Award I2301E0026), administered by A*STAR, as well as supported by Alibaba Group and NTU Singapore through Alibaba-NTU Global e-Sustainability CorpLab (ANGEL).

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

# A. More Quantitative Analysis

To further validate the effectiveness of our *Generative Point Cloud Registration* paradigm, we integrate the handcrafted geometric descriptor, FPFH, into our framework, forming **Generative FPFH**. As shown in Table 4, this generative variant achieves a significant performance improvement over the baseline FPFH on the 3DMatch benchmark dataset, regardless of whether DINOv2 encoding or Stable-Diffusion encoding is used.

*Table 4.* Comparison of the methods on rotation, translation, and Chamfer distance on **3DMatch** (Zeng et al., 2017) benchmark dataset.

| | Rotation (*deg*) | | | | | Translation (*cm*) | | | | | Chamfer (*mm*) | | | | |
|---|---|---|---|---|---|---|---|---|---|---|---|---|---|---|---|
| | Accuracy ↑ | | | Error↓ | | Accuracy ↑ | | | Error↓ | | Accuracy ↑ | | | Error↓ | |
| Methods | 5 | 10 | 45 | Mean | Med. | 5 | 10 | 25 | Mean | Med. | 1 | 5 | 10 | Mean | Med. |
| FPFH (Rusu et al., 2009) | 69.1 | 82.9 | 91.2 | 15.0 | 3.1 | 25.8 | 53.9 | 75.1 | 37.4 | 9.1 | 52.5 | 74.2 | 79.2 | 57.6 | 0.9 |
| Generative FPFH$^{\text{DINOv2}}$ | 83.7 | 91.4 | 95.9 | 7.9 | 2.1 | 33.9 | 64.0 | 80.5 | 24.7 | 6.9 | 62.7 | 80.6 | 83.2 | 50.0 | **0.6** |
| Generative FPFH$^{\text{SD}}$ | **88.2** | **94.7** | **96.9** | **6.1** | **2.0** | **37.6** | **68.9** | **85.5** | **18.5** | 6.6 | **66.9** | **85.7** | **88.8** | **41.2** | 0.6 |
| *Improvement* ↑ | *19.1* | *11.8* | *5.7* | *8.9* | *1.1* | *11.8* | *15.0* | *10.4* | *18.9* | *2.5* | *14.4* | *11.5* | *9.6* | *16.4* | *0.3* |

# B. More Visualization Results of Match-ControlNet

In Fig.7, we present additional visualization results of the RGB image pairs generated by our *Match-ControlNet*, along with the corresponding colorized source and target point clouds. Furthermore, Fig.8 visualizes the image pairs generated by our zero-shot *Match-ControlNet*, while Fig. 9 showcases image pairs produced by the finetuned *Match-ControlNet*.

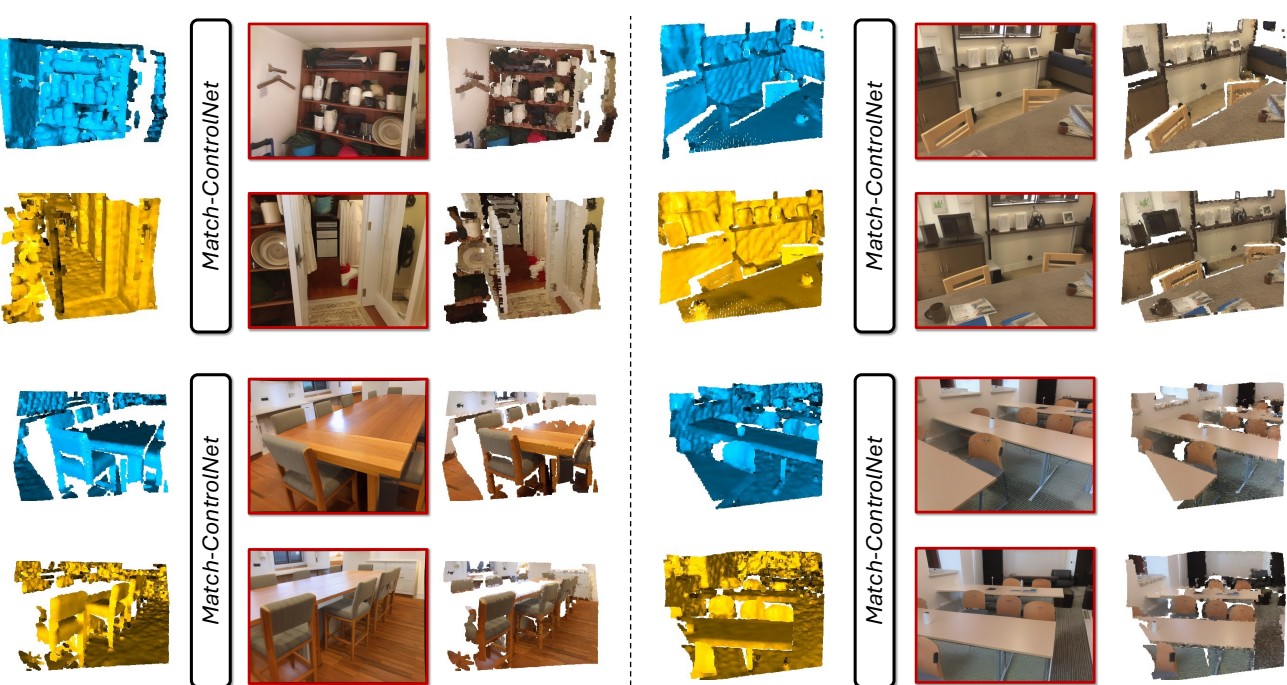

*Figure 7.* More visualization results of the generated RGB image pairs and the formed color source and target point clouds.

# Zero-Shot Match-ControlNet Generation

*Figure 8.* Source and target image generation via **zero-shot** *Match-ControlNet* without any finetuning.

**Finetuned Match-ControlNet Generation**

*Figure 9.* Source and target image generation via **finetuned** *Match-ControlNet*.

