# OpenReview forum: "Generative Point Cloud Registration"
_ICML.cc/2025/Conference — ICML 2025 poster_

### Official Review · Reviewer_P29t · 2025-03-10

**Overall Recommendation:** 4

**Summary:**

This work introduces a novel method for point cloud registration that aims to generate geometry-consistent RGB image pairs from paired point sets. These generated RGB image pairs are then used to enhance the performance of point-based registration methods. The proposed approach incorporates two key innovations: a newly designed coupled conditional denoising technique and a prompt guidance mechanism, both of which are developed to enforce cross-view texture consistency. Extensive experiments conducted on the ScanNet and 3DMatch datasets demonstrate the effectiveness of the proposed method.

**Claims And Evidence:**

The claimed contributions are well-supported by the proposed method and experimental results.

**Essential References Not Discussed:**

No. The literature review looks comprehensive.

**Experimental Designs Or Analyses:**

experimental designs or analyses are solid.

**Methods And Evaluation Criteria:**

The newly designed coupled conditional denoising technique and prompt guidance mechanism are technically sound and interesting. The evaluation metrics are reasonable.

**Other Comments Or Suggestions:**

No

**Other Strengths And Weaknesses:**

Strengths:
The idea of using generated image pairs to enhance the registration performance of point-based models is highly innovative. Both quantitative and qualitative results demonstrate the effectiveness of the proposed method, showcasing its potential to improve registration accuracy.

Weaknesses:
1. Point set registration encompasses both rigid and non-rigid transformations. However, this paper focuses exclusively on rigid transformations. This should be clearly stated early in the introduction to set appropriate expectations for readers.
2. In Table 2, the proposed method shows a noticeable improvement in mean errors but no significant enhancement in median error. This discrepancy warrants further explanation or analysis to clarify the underlying reasons.
3. The baseline models primarily consist of older methods published several years ago. To better highlight the advancements of the proposed method, it would be beneficial to include evaluations against more recent state-of-the-art models.
4. There is a concern that ControlNet might have been trained on datasets such as ScanNet or similar datasets, which could reduce the difficulty of the generation task. It would be valuable to verify whether the proposed method remains effective on more challenging datasets to ensure its generalizability.

**Questions For Authors:**

See the weakness section.

**Relation To Broader Scientific Literature:**

The proposed method can serve as a plug-and-play module for various existing point-based registration models.

**Theoretical Claims:**

No theoretical proof in the paper.

---

> ### Author Rebuttal · Authors · 2025-03-31
>
> **Q1: Clarification on rigid transformations.**
> **A1:** Thank you for your valuable suggestion. In the revised introduction, we will explicitly state that our work focuses exclusively on the rigid point cloud registration problem to set clear expectations for the readers.
>
> **Q2: The discrepancy between the noticeable improvement in mean errors and the lack of significant enhancement in median error in Table 2 requires further explanation.**
> **A2:** Thank you for the insightful comment. We clarify that this discrepancy arises because mean and median errors capture different aspects of performance. Specifically, the mean error reflects the overall precision and is sensitive to outliers. The noticeable improvement in mean error stems from a substantial reduction in large-error cases, indicating that our method can effectively handle challenging scenarios. Instead, the median error reflects the model’s performance on the majority of cases, which was already strong and thus shows minimal change. We will clarify this distinction in the revised manuscript to avoid confusion and better highlight the strengths of our approach.
>
> **Q3: Comparison with newer state-of-the-art.**
> **A3:** We appreciate the reviewer’s valuable suggestion. Following the reviewer’s recommendation, we have conducted an additional comparison with a recent state-of-the-art method, PARE-Net (ECCV 2024), and included the results in the table below. The new results show that our proposed Generative FCGF (SD) consistently outperforms PARE-Net, confirming our superior registration performance. We will include these comparisons in our revised version.
>
> | Methods               | Rot@5  | Rot@10 | Rot@45 | Mean | Med. | Trans@5 | Trans@10 | Trans@25 | Mean | Med. |
> |-----------------------|------------------|----------|----------|---------------|---------------|----------------------|----------|----------|---------------|---------------|
> | PARE-Net (ECCV'2024)                | 75.6             | 82.1     | 86.5 | 21.0          | 2.1           | 40.6                 | 63.5     | 77.4     | 47.4          | 6.4          |
> | Generative FCGF  | **94.3**         | **96.7** |  **98.1**     | **4.5**        | **1.4**        | **54.3**              | **81.5** | **93.1** | **12.5**        | **4.7**         |
>
> **Q4: The concern is that ControlNet may have been trained on easier datasets like ScanNet, which could simplify the generation task. It's important to test the proposed method on more challenging datasets to confirm its generalizability.**
> **A4:** Thank you for the thoughtful comment. **(i)** We would like to clarify that, although the training dataset for the depth-conditioned ControlNet model has not been publicly released, the depth maps used in that model were generated by the MiDaS depth estimation model. By contrast, our experiments utilize real depth maps captured directly by depth sensors. Despite the significant domain gap between the MiDaS-estimated and sensor-captured depth maps, our method still achieves impressive generation quality, highlighting its robustness and generalization capability; **(ii)** To further validate the generalizability and robustness of our Match-ControlNet in more challenging scenarios, we conducted an additional experiment where we casually captured low-overlap photos of a cluttered, unconstrained indoor environment (i.e., the author's room) using a mobile phone. We then estimated their depth maps using DUSt3R for subsequent Match-ControlNet generation (without any model fine-tuning). The figure (https://anonymous.4open.science/r/rebuttal-688D/wild_vis.pdf) demonstrates that, even under these challenging in-the-wild conditions with a different depth-map source, our method still achieves impressive cross-view consistency and generation quality. These results confirm the practical effectiveness and excellent generalizability of our approach.

---

> > ### Comment · Reviewer_P29t · 2025-04-07
> >
> > I thank the authors for their comprehensive and detailed responses to my questions. After reading their rebuttal, I believe the authors have addressed my concerns. I believe the manuscript now meets the acceptance standards, and I will maintain my score.

---

### Official Review · Reviewer_z9of · 2025-03-13

**Overall Recommendation:** 3

**Summary:**

The paper proposes a new perspective on Point Cloud Registration: Generative Point Cloud Registration. Compared to traditional methods or purely geometry-based learning methods, the paper incorporates image generative models. The input is a point cloud pair with unknown pose, and the output is the transformation between them.

Specifically, the proposed Match-ControlNet generates corresponding images for each point cloud, and then uses image features as additional information to assist existing point cloud matching methods in finding better correspondences.

The authors design Match-ControlNet's condition images and prompts to ensure geometric and texture consistency. The authors add this method on top of multiple baselines and experimentally demonstrate that Generative Point Cloud Registration can improve the performance of baselines.

## update after rebuttal
I carefully reviewed the materials provided by the author during the rebuttal phase. With these additional details, I believe the experimental integrity of the paper has improved. Initially, I had concerns about the completeness of the experimental section. However, after the author's explanations and provision of additional materials, my concerns have been adequately addressed, justifying a higher rating for the manuscript.

**Claims And Evidence:**

Yes, I believe introducing image information would be helpful for point cloud matching.

**Essential References Not Discussed:**

None

**Experimental Designs Or Analyses:**

1. I remember ScanNet has images, and the experiment lacks a comparison between the improvement of using real multi-view consistent images as additional information over the baseline and the magnitude of improvement using generated images over the baseline.
2. For methods like DUSt3R, although not used for point cloud registration problems, they can obtain the relative pose between two viewpoints through images alone. I'm curious about how the proposed method would compare with these methods? In other words, when both images and point clouds are available, how important are the point cloud features for the registration task?
3. I think the authors should add information about how different image generation results affect the estimated pose for the same pair of point clouds, or provide the mean and variance of improvements over the baseline after multiple executions.

**Methods And Evaluation Criteria:**

I see methodological flaws in the proposed approach. While the authors introduced requirements like 2D-3D Geometric Consistency and Cross-view Texture Consistency, the method essentially processes a stack of images using ControlNet. This alone does not robustly ensure multi-view consistency. Achieving consistency across views fundamentally requires accurately modeling the joint distribution of multiple images—a key aspect seemingly overlooked here. Instead, the current method relies on fine-tuning ControlNet on general image diffusion models, yet fails to introduce explicit constraints to enforce consistency. As a result, I remain doubtful about the reliability of the generated multi-view outcomes.

Furthermore, since the approach presented in the paper often serves as a workaround in various image diffusion-based applications, it exhibits certain instability.

For instance, works like Zero123++ accomplish multi-view information fusion by implementing novel modules within attention mechanisms—elements conspicuously missing from this paper.

**Other Comments Or Suggestions:**

I hope to show more visualization results of point cloud overlap after matching, rather than just the images generated by ControNet.

**Other Strengths And Weaknesses:**

This paper combines the generative capabilities of ControlNet with Point Cloud Registration in an application-oriented article. Its strength lies in providing a new perspective on the Registration problem, using generative methods to compensate for the lack of image-point cloud pairs. Its weakness is that the proposed method resembles more of an experimental report, and neither theoretically nor through rigorous experimental analysis can it explain why this approach enables ControlNet to achieve multi-view texture consistency.

**Questions For Authors:**

None

**Relation To Broader Scientific Literature:**

- DUSt3R: Geometric 3D Vision Made Easy[CVPR2024]
    - This paper discusses the problem of pose estimation using only images
- Zero-1-to-3: Zero-shot One Image to 3D Object[ICCV2023]
- Zero123++: a Single Image to Consistent Multi-view Diffusion Base Model
    - The above two papers discuss how to better inject multi-view consistency into Diffusion

**Theoretical Claims:**

No Theoretical Claims

---

> ### Author Rebuttal · Authors · 2025-03-31
>
> **Q1: Lack of joint distribution modeling of multi-view images?**
> **A1:** We respectfully clarify that our coupled denoising mechanism has implicitly modeled the joint distribution of multi-view images (i.e., cross-view images in our task).
> Formally, the likelihood over the cross-view image pair can be expressed as $\mathbb{E}[p_\theta({x}^{P}, {x}^Q)]$ (we omit conditional variables for simplicity).
> By concatenating the cross-view images into a unified variable
> ${x}^{PQ} = Cat([{x}^P, x^Q])$, we can model the joint distribution of cross-view images via the likelihood over the concatenated representation: $\mathbb{E}[p_\theta({x}^{PQ})]$. As such, we can derive a variational lower bound on the data log-likelihood for optimization: L=E_q [log p_θ(x_0^{PQ} | x_1^{PQ}) - ∑_{t>1} D_KL(q(x_{t-1}^{PQ} | x_t^{PQ}, x_0^{PQ}) || p_θ(x_{t-1}^{PQ} | x_t^{PQ})) - D_KL(q(x_T^{PQ}| x_0^{PQ}) || p(x_T^{PQ}) )].
>
> Through the reparameterization trick, maximizing this lower bound is equivalent to minimizing our training loss in Eq. 5. This derivation demonstrates that our coupled denoising implicitly models the joint distribution of cross-view images, thereby effectively enforcing cross-view consistency generation. We will clarify this in our revision.
>
> **Q2: The proposed method is a workaround and exhibits instability? Absence of explicit attention-based modules like Zero123++?**
> **A2:**
> We respectfully clarify that our method is not a workaround, but an efficient and elegant design that enables effective cross-view information fusion and consistency learning with minimal overhead. Unlike methods like Zero123++, which introduce an additional complex attention modules for fusion, our attention design is native. We employ a novel coupled denoising mechanism that naturally extends the inherent intra-image self-attention to both inter- and intra-image attention, without any architectural modifications (see Eq. 4). We note that this design effectively unlocks the zero-shot consistency generation ability of Stable Diffusion, acquired through extensive large-scale pretraining, thereby enhancing both generation quality and cross-view consistency. As evidenced by extensive experiments, our method consistently delivers stable results across diverse scenarios (see Table 5, 6, and Fig. 5). Moreover, we tested on low-overlap, in-the-wild data (from the author's room) captured by a mobile phone. The results (https://anonymous.4open.science/r/rebuttal-688D/wild_vis.pdf) further confirm its stability in unconstrained conditions (please see A4@P29t for details).
>
> **Q3: Real vs. generated images comparison.**
> **A3:** As shown in Table 3 (third block) in our manuscript, we had compared ColorPCR using ground-truth images with our generative ColorPCR that utilizes generated images. Surprisingly, our generated images can even achieve superior performance across most metrics. We attribute this improvement to that our generated images effectively mitigate calibration errors and lighting challenges inherent in real-world data, as illustrated in Fig. 6.
>
> **Q4: Comparison with image-only methods (e.g., DUSt3R).**
> **A4:** We would like to emphasize that our paper targets the point cloud matching rather than the image matching. Therefore, comparisons with image-only methods actually fall outside the scope of our research. Below, we report the score of SOTA image-only method, DUSt3R, on 3DMatch. It shows that, due to the absence of geometric and scale information inherent in point clouds, DUSt3R demonstrates limited precision, highlighting the importance of point cloud features for accurate registration.
>
> | Methods               | Rot@5  | Rot@10 | Rot@45 | Mean | Med. | Trans@5 | Trans@10 | Trans@25 | Mean | Med. |
> |-----------------------|------------------|----------|----------|---------------|---------------|----------------------|----------|----------|---------------|---------------|
> | DUSt3R                | 50.9             | 64.2     | **98.5** | 10.0          | 4.9           | 6.6                  | 21.2     | 61.7     | 23.3          | 19.7          |
> | Generative FCGF  | **94.3**         | **96.7** | 98.1     | **4.5**        | **1.4**        | **54.3**              | **81.5** | **93.1** | **12.5**        | **4.7**         |
>
> **Q5: Variance analysis across generated images.**
> **A5:** Thank you for your suggestion. We repeated the evaluations on 3DMatch using varying random seeds (123, 1234, 12345) for Match-ControlNet. The mean and variance of Generative FCGF are 93.7 ± 0.6 for Rot@5 and 54.1 ± 0.1 for Trans@5, which still exhibits a significant performance gain over the baseline, validating the robustness of our method.
>
> **Q6: More visualization results of point cloud overlap after matching.**
> **A6:** We provide more registration visualisation results in figure (https://anonymous.4open.science/r/rebuttal-688D/reg_vis.pdf), qualitatively showing our excellent precision. We will include them into our revised version.

---

> > ### Comment · Reviewer_z9of · 2025-04-02
> >
> > I appreciate the author's further explanation of the novelty of the method and the additional experimental results provided, but unfortunately, I was unable to open or verify any of the PDFs linked anonymously by the author. If others can offer a way to access these PDFs or if the author can provide clearer evidence regarding the supplementary results, I am willing to adjust my score based on those results.

---

> > > ### Author Response · Authors · 2025-04-02
> > >
> > > Thank you very much for your response and for your willingness to reconsider the score based on the supplementary results. We sincerely apologize for the inconvenience caused by the instability of the anonymous GitHub system.
> > >
> > > To access the supplementary PDFs, reviewers can click the “Download Repository” button on the linked anonymous webpage. Alternatively, the direct download button link is: "https://anonymous.4open.science/api/repo/rebuttal-688D/zip", which contains all supplementary PDFs for review.
> > >
> > > Additionally, to further ensure accessibility, we have provided an anonymous Google Drive folder containing the same supplementary PDFs:
> > > "https://drive.google.com/drive/folders/1cVcv5Nw8eNUhgNaHsIf6LEJ8MBUmRa32".
> > >
> > > We truly appreciate your time and consideration. If there are still any access issues, please don’t hesitate to let us know.

---

### Official Review · Reviewer_evgg · 2025-03-13

**Overall Recommendation:** 4

**Summary:**

This paper introduces a novel approach to point cloud registration by leveraging generative models to synthesize 2D images from 3D point clouds, enabling better feature extraction and matching for registration tasks. Traditional methods primarily rely on 3D feature matching, which often struggles in scenarios with low overlap, noise, and incomplete data. The authors propose Match-ControlNet, a framework that utilizes generative 2D diffusion models (such as Stable Diffusion) to improve geometric and texture consistency for robust point cloud alignment.

**Claims And Evidence:**

Yes

**Essential References Not Discussed:**

No

**Experimental Designs Or Analyses:**

Yes

**Methods And Evaluation Criteria:**

Yes

**Other Comments Or Suggestions:**

None

**Other Strengths And Weaknesses:**

Strengths:

$\cdot$ Novel Paradigm: The paper introduces a new approach to point cloud registration by leveraging generative 2D models to enhance 3D matching tasks, bridging a gap between 2D and 3D data processing.

$\cdot$ Match-ControlNet: The proposed Match-ControlNet improves geometric and texture consistency between generated image pairs, which aids in better point cloud registration.

$\cdot$ Enhanced Feature Fusion: The work integrates both zero-shot geometric-color feature fusion and XYZ-RGB fusion, providing additional visual cues for more accurate correspondence estimation.

$\cdot$ Generalization and Plug-and-Play Nature: The framework can be integrated with various 3D registration methods without requiring significant modifications.

$\cdot$ Empirical Validation: Extensive experiments on benchmark datasets (3DMatch, ScanNet) demonstrate improved registration accuracy compared to existing methods.

$\cdot$ Addressing Low-Overlap Issues: The proposed method shows strong performance in challenging cases with low overlap and noisy point clouds.


Weaknesses:

$\cdot$ Dependency on Generative Models: The approach heavily relies on the quality of the generated 2D images, which could introduce artifacts or inconsistencies in certain scenarios.

$\cdot$ Computational Overhead: Generating 2D images using generative models such as Stable Diffusion and performing additional feature fusion may introduce extra computation costs.

$\cdot$ Few-Shot Fine-tuning Requirement: While the method offers a zero-shot solution, performance improvements through fine-tuning indicate that additional labeled data might still be necessary for optimal results.

$\cdot$ Potential Sensitivity to Viewpoint Selection: The quality of the generated images depends on the viewpoint chosen for rendering, which might impact registration performance in complex 3D scenes.

$\cdot$ Limited Real-World Evaluation: The datasets used (3DMatch, ScanNet) are standard benchmarks, but real-world performance in applications like autonomous driving or robotics remains to be seen.

Overall, the paper presents a promising direction for improving point cloud registration by incorporating generative models, though computational efficiency and generalization to diverse real-world scenarios might require further investigation.

**Questions For Authors:**

None

**Relation To Broader Scientific Literature:**

The key contributions of Generative Point Cloud Registration are closely related to several areas in the broader scientific literature, particularly in 3D vision, generative modeling, and point cloud processing.

**Theoretical Claims:**

Yes

---

> ### Author Rebuttal · Authors · 2025-03-31
>
> **Q1: The approach heavily relies on the quality of the generated 2D images, which could introduce artifacts or inconsistencies in certain scenarios?**
> **A1:** The image generation quality is indeed crucial to the overall performance. Notably, our Match-ControlNet successfully unlocks the generalizable zero-shot consistency generation ability inherently learned through the large-scale pretraining of Stable Diffusion and ControlNet (see Sec.3.2 and Sec.3.3).
> This large-scale pretraining on consistency generation can effectively help mitigate potential artifacts and ensures strong consistency across a wide range of scenarios. On top of that, our coupled denoising, coupled prompt guidance, and few-shot consistency fine-tuning strategy further enhance generation reliability. These components are extensively validated through comprehensive real-world benchmark experiments, especially in challenging scenarios with low overlap, occlusions, and cluttered environments.
>
> **Q2: About computational overhead.**
> **A2:** While image generation and feature fusion introduce additional computational cost, they substantially enhance the quality  of geometric descriptors, leading to significantly improved registration robustness. To better balance performance and efficiency, our future work will explore single-step denoising techniques and knowledge distillation mechanisms to accelerate both image generation and feature fusion.
>
> **Q3: Few-shot fine-tuning requirement.**
> **A3:** Our method already demonstrates impressive performance in zero-shot scenarios (as shown in Fig. 8). The few-shot fine-tuning, requiring only a minimal amount of image pairs (~3K samples), further improves accuracy. We believe that this lightweight requirement is both practical and consistent with common industry practices.
>
> **Q4: Potential sensitivity to viewpoint selection?**
> **A4:** We thank the reviewer for raising this important point. In practice, our method exhibits strong robustness to viewpoint selection. Notably, our extensive experiments on the ScanNet and 3DMatch benchmarks already cover a wide range of diverse and challenging viewpoint configurations. Across these varying viewpoint conditions, our method consistently achieves significant improvements in registration performance, demonstrating its reliability under viewpoint selection. We attribute this robustness to the powerful generalization ability of Match-ControlNet, which benefits from the large-scale pretraining of foundation models, as well as our coupled conditional denoising design. We will incorporate this discussion into the revised manuscript.
>
> **Q5: About real-world applicability in autonomous driving or robotics.**
> **A5:** We appreciate the reviewer’s concern.
> **(i)** While 3DMatch and ScanNet are standard benchmarks, it is important to note that they are constructed from real-world RGB-D scans and have been widely adopted in robotics and embodied AI research, especially ScanNet, which is a common benchmark for indoor robotic perception;
> **(ii)** Furthermore, these datasets capture realistic and challenging conditions, such as low overlap, occlusions, and cluttered layouts, which are highly representative of deployment scenarios in indoor robotics;
> **(iii)** Regarding autonomous driving scenarios, as shown in the figure (https://anonymous.4open.science/r/rebuttal-688D/outdoor_vis.pdf), our Match-ControlNet can generate cross-view consistent images from partial LiDAR scan-based sparse depth maps, demonstrating promising generation effectiveness in outdoor self-driving scenes (please refer to A1@synA for more details);
> **(iv)** Moreover, we tested our method on in-the-wild, low-overlap data (from the author's room) captured by a mobile phone. The resulting generations (https://anonymous.4open.science/r/rebuttal-688D/wild_vis.pdf) further confirm the robustness of our approach in the real-world, unconstrained environment (please refer to A4@P29t for more details). We will further expand the discussion on applications to autonomous driving and robotics in our revision.

---

> > ### Comment · Reviewer_evgg · 2025-04-01
> >
> > Thanks for your response. My concerns have been addressed. I keep my score.

---

> > > ### Author Response · Authors · 2025-04-02
> > >
> > > Thank you for your response. We're happy to hear that our clarification resolved your concerns, and we appreciate your time and effort in reviewing our paper.

---

### Official Review · Reviewer_synA · 2025-03-18

**Overall Recommendation:** 3

**Summary:**

This paper proposes a new 3D Registration method, Generative Point Cloud Registration, which connects advanced 2D generative models with 3D matching tasks to improve registration performance. The key idea in this paper is to generate cross-view consistent image pairs that are well aligned with source and target point clouds, so as to achieve geometric-color feature fusion to promote robust matching. Experimental results show that it can be seamlessly integrated into various registration methods to enhance their performance.

**Claims And Evidence:**

Yes

**Essential References Not Discussed:**

Also starting from the generative model registration problem, this paper lacks references to FREEREG (ICLR24).

[1] Wang H, Liu Y, Wang B, et al. Freereg: Image-to-point cloud registration leveraging pretrained diffusion models and monocular depth estimators[J]. arXiv preprint arXiv:2310.03420, 2023.

**Experimental Designs Or Analyses:**

The experimental setting was adequate and reasonable, and the only deficiency was that the analysis was not carried out in the outdoor scene, regardless of whether the results were good or bad.

**Methods And Evaluation Criteria:**

The proposed method is of great significance in the field of point cloud registration, and an innovative idea of point cloud registration is proposed.

**Other Comments Or Suggestions:**

N/A

**Other Strengths And Weaknesses:**

Strengths:

1. The experimental results are excellent, demonstrating strong performance on both the 3DMatch and ScanNet datasets.

2. The writing of the paper is well-executed, and the content is easy to understand.

3. This paper has some innovation. From the perspective of generation, it provides a new idea for point cloud registration field. But from another point of view, there are many methods to enhance the point cloud registration effect through the texture and color information of 2D images, such as Colorpcr (CVPR24), PointMBF (CVPR24), PEAL (CVPR23).

Weaknesses:

1. The applicability of the method in this paper is poor, and I personally understand that the method cannot be applied to outdoor LiDAR scenes. Because this paper generates 2D images from depth maps. In the follow-up, we can add some experiments to verify whether it is effective for outdoor scenes, and consider whether this problem can be solved in future work.

2. No code is provided for verification.

**Questions For Authors:**

I have a small question, as shown in Figure 5, how to ensure that the same texture and color information is generated in the overlapping area when faced with low overlap? If this point cannot be guaranteed, I think performance should not improve significantly or even decrease at low overlap.

**Relation To Broader Scientific Literature:**

This paper presents a new idea in the field of point cloud registration. The effect of point cloud registration is enhanced by generative method.

**Theoretical Claims:**

There is no Theoretical Claim.

---

> ### Author Rebuttal · Authors · 2025-03-31
>
> **Q1: Discussion on applicability in outdoor LiDAR scenes.**
> **A1:** We sincerely appreciate this insightful comment. Our current Match-ControlNet indeed targets leveraging depth maps rather than LiDAR data for image generation. Compared to forward-facing depth maps, outdoor LiDAR point clouds provide omnidirectional (360-degree) scans that cannot be directly represented as conventional single-viewpoint depth maps due to their inherent multi-directional nature. In this rebuttal, we demonstrate the feasibility of partially projecting LiDAR points from predefined viewpoints to produce sparse depth maps for Match-ControlNet generation, yielding promising preliminary generation results as illustrated in the figures (https://anonymous.4open.science/r/rebuttal-688D/outdoor_vis.pdf). For future work, we plan to comprehensively address omnidirectional scans by employing multi-view depth maps or equirectangular images (a 2D image representation of LiDAR data) to adapt our Match-ControlNet to LiDAR point clouds. We will include the above discussion into our revised manuscript.
>
> **Q2: Consistency of texture information under low overlap?**
> **A2:** Our texture consistency under low overlap can be largely ensured by following four aspects: **(i) Extensive pre-training knowledge:** Our Match-ControlNet effectively unlocks the intrinsic zero-shot texture consistency capabilities of foundation models, which are derived from large-scale data pre-training. We believe this extensive training provides a strong foundation for ensuring texture consistency across diverse real-world challenges, including low-overlap scenarios.
>  **(ii) Joint image-level and prompt-level texture consistency enhancement:** Our coupled conditional denoising mechanism and the coupled prompt guidance jointly prompt the texture consistency by incorporating both image-level texture consistency interaction and the prompt-level texture consistency guidance, significantly enhancing the texture coherence in low-overlap settings; **(iii) Effective texture consistency fine-tuning mechanism:** We introduce a few-shot consistency fine-tuning strategy (see Sec. 3.4) that requires only a small number of samples to further enhance the generation robustness of Match-ControlNet, particularly in challenging scenarios;
> **(iv) Extensive validation under low-overlap conditions:** To validate the robustness of our method under low-overlap conditions, we significantly increased the viewpoint separation (from 20 to 50 degrees on ScanNet and from 20 to 40 degrees on 3DMatch) as detailed in Line 311 (right column) and Line 362 (left column). Both quantitative and qualitative results (Fig. 7, first image; Fig. 8, third column) demonstrate reliable texture consistency. Additionally, we tested our method on in-the-wild, low-overlap data captured by a mobile phone. The resulting generations (https://anonymous.4open.science/r/rebuttal-688D/wild_vis.pdf) further confirm the robustness of our approach (please refer to A4@P29t for more details).
>
>
> **Q3: Reference Issue.**
> **A3:**  We thank the reviewer for pointing out the missing FREEREG reference. We will include and thoroughly discuss this work in the revised manuscript.
>
> **Q4: Code availability.**
> **A4:**  We will release our implementation publicly upon acceptance to facilitate reproducibility and enable broader verification by the community.

---

> > ### Comment · Reviewer_synA · 2025-04-05
> >
> > Thanks for your response. My concerns have been addressed. I will raise the score appropriately.

---

> > > ### Author Response · Authors · 2025-04-05
> > >
> > > Thank you very much for your positive comments. We’re glad to hear that our response helped address your concerns. We truly appreciate the time and effort you put into reviewing both our paper and our rebuttal, and we’re especially grateful for your willingness to reconsider the score.

---

### Decision · Program_Chairs · 2025-05-01

**Decision:**

Accept (poster)

**Comment:**

All the reviewers agree that the paper provided novel perspective for addressing 3D point clouds registration problem, the method is interesting and outperforms various state-of-the-art. During rebuttal, more clarification and explanations and additional experimental results have been provided. All the reviewers are convinced that the paper has made important contribution to the field. After the interactions with the authors, two reviewers raised their overall recommendation scores.